# Trends in Mental Wellbeing of US Children, 2019–2022: Erosion of Mental Health Continued in 2022

**DOI:** 10.3390/ijerph21020132

**Published:** 2024-01-25

**Authors:** Neeraj Bhandari, Shivani Gupta

**Affiliations:** 1Department of Healthcare Administration and Policy, School of Public Health, University of Nevada, Las Vegas, NV 89154, USA; 2Department of Health Administration, College of Business, University of Houston-Clear Lake, Houston, TX 77058, USA; gupta@uhcl.edu

**Keywords:** mental health, children, COVID-19

## Abstract

We provide fresh estimates of a change in the nationwide prevalence of mental health symptoms among US children during the COVID-19 pandemic using National Health Interview Survey data (2019–22) on children aged 2–17 years (n = 27,378; age subgroups 2–5, 6–11, and 12–17) to assess overall mental distress and 19 specific outcomes related to developmental, communicative, cognitive, affective, and behavioral domains. Raw and adjusted (for socio-demographics) linear regressions estimated the change in prevalence for each outcome between 2019 (baseline year) and three succeeding years (2020–2022). Summary scores for mental distress rose between 2019 and 2020 (1.01 to 1.18 points, range of 0–15), declined slightly in 2021 (1.09), and climbed sharply again in 2022 (1.25). The declines primarily affected adolescents (1.11 at baseline, 1.24 in 2020, 1.30 in 2021, and 1.49 in 2022). Specific outcomes belonging to all domains of mental health showed similar increases in prevalence. US children suffered significant erosion of mental health during the COVID-19 pandemic that continued into 2022. Expansion of mental health programs aimed at school-going children will likely be needed to respond effectively to the ongoing crisis.

## 1. Introduction

The SARS-CoV-2 pandemic ushered several social (e.g., the decline in social mobility and peer interaction) and policy responses (e.g., lockdowns and school closures) that upended long-standing patterns of social and cultural life worldwide and intensified population-wide exposure to potent psychopathic stressors [1,2,3]. The ensuing disruptions contributed to the rise in the prevalence of mental health problems in children [4,5], but the precise extent and durability of the problem are subject to intensifying debate [6,7,8]. Early studies suggested steep declines in mental health but were based mainly on cross-sectional or non-representative data [9,10]. Longitudinal cohort studies have yielded more mixed results [6,7,11,12,13,14,15]. For instance, a recent large meta-analysis of 53 longitudinal cohort studies by Madigan et al. (2023) found significant increases in depression and anxiety during the pandemic compared to pre-pandemic levels, particularly in adolescent girls and children belonging to mid-to-high income families [6]. The strengths of this review were a large cumulative sample size and a strong representation of pre-pandemic and pandemic time periods. Another study that combined data from 12 longitudinal studies (most of them US-based) found significant elevations in depression but no change in anxiety [15]. On the other hand, a large multicenter longitudinal study of 4702 children between 9 and 12 years old found only minor changes in depressive and other affective symptoms, mostly among children of parents with less supportive parenting styles [7].

The existing literature presents several limitations for ascertaining the precise burden of mental illness. First, there is a scarcity of prevalence studies that use data representative of large national populations like the United States. Most studies, including longitudinal cohort studies, have small sample sizes or rely on non-representative sampling strategies (such as online surveys) that have documented problems in data collection. Second, many studies used surveys that were restricted in geographic scope to sub-national regions such as states or cities. Furthermore, no large nationwide study has reported data from 2022, with most covering only the initial periods of the pandemic. Finally, many studies have restricted focus on one or a few aspects of mental health and rarely explore the full spectrum of mental health changes affecting key dimensions of mental wellbeing. Our study seeks to address these limitations by providing fresh estimates of change in the nationwide prevalence of US children’s mental health symptoms between 2019 (baseline year) and three succeeding years (2020–2022), both in terms of the overall burden of mental illness and several key metrics related to developmental, communicative, cognitive, affective, and behavioral domains.

## 2. Methods

### 2.1. Data, Setting, and Population

National Health Interview Survey (NHIS) annually collects information regarding health, healthcare use, access to providers, insurance, and other related topics from a nationally representative sample of non-institutionalized adult and child respondents through detailed in-person interviews. The survey universe includes residents of households and all noninstitutional group living units (e.g., homeless shelters, rooming houses, and group homes) but excludes people with no permanent household address (e.g., out-of-shelter homeless); active-duty military personnel and civilians residing on military bases; residents of long-term care institutions (e.g., nursing homes); persons in prisons, juvenile detention centers, and halfway houses; and U.S. nationals living abroad. NHIS collects data continuously from January to December and employs geographically clustered sampling techniques to select the sample of target households. Each year, a new sample of respondents is interviewed. Trained interviewers conduct face-to-face interviews in each respondent’s home with one randomly selected adult and child, but follow-ups to complete interviews sometimes require contact over the telephone. The interviews are conducted through computer-assisted personal interviewing (CAPI) using Blaise computer software that presents questions on computer screens to each interviewer [16]. NHIS has long been regarded as a gold standard for survey collection methodology and has had historically high response rates. Survey response rate for each study year was as follows: 59.1% (2019), 47.8% (2020), 49.9% (2021), and 47.7% (2022). We used four years of NHIS data (2019–22) on children aged 2–17 years (n = 27,378) to generate three age subgroups: 2–5 (n = 6222), 5–11 (n = 9298), and 11–17 (n = 11,858). Our stratification into distinct age subgroups reflects an acknowledgment of distinctive stages of child social and cognitive development that are linked to relatively unique set of vulnerabilities to stressors in child development [17]. The University of Nevada, Las Vegas Institutional Review Board determined that this study did not require IRB review because it used existing, anonymized, publicly available data.

### 2.2. Outcome Variables

All our outcome measures are based on survey items that asked parents to report on whether their child currently had a specific diagnosis or symptoms related to each dimension of mental health. We generated five variables measuring parent-reported prevalence of developmental and learning difficulties for all age subgroups using survey items that asked whether child was reported by a health professional to currently have an Attention-Deficit/Hyperactivity Disorder (ADHD) or Attention-Deficit Disorder (ADD), intellectual disability, autism, developmental delay, or a learning difficulty. We measured the prevalence of impaired communication from separate survey items for children aged 2–4 and 5–17 that were worded slightly differently. For children aged 2–4, we dichotomized the variables to assess prevalence of any difficulty faced by the child in understanding the adult respondent, or, conversely, by respondents in understanding the child. For children aged 5–17, we assessed whether the child had any difficulty being understood by people inside or outside their household. We dichotomized four survey items to capture any difficulty with learning (age 2–17), remembering things (age 5–17), and whether child seemed anxious and depressed at least every week (age 5–17). Behavioral difficulties were measured by 6 items soliciting information about any difficulties with controlling behavior and concentrating (age 2–4), and difficulties in playing, violence toward peers, handling disruptions in routine, and making friends (age 5–17). Finally, we sought to quantify each child’s total burden of mental distress (affecting all dimensions of mental health) by coding a summary variable that computed a non-weighted sum of all dichotomized variables.

### 2.3. Control Variables

In order to account for changes in sample composition (e.g., socio-demographics) across study years, we generated control variables for age; sex; race/ethnicity; household size; self-ratings of health (1 = at least good health); type of insurance coverage (private, Medicaid/public, other, uninsured); family structure (single parent, married two-parent, cohabiting two-parent, no parent); family race composition (same race, mixed race, unknown/no parents); household income groups (based on federal poverty line), parental age, employment, and education (at least one parent has college degree); family food security (high = 1); residential arrangement (owned house = 1); and presence of at least one elderly person living with family

### 2.4. Analysis

We generated summary statistics for full sample and all age subgroups by computing means (continuous and discrete variables) and proportions (binary variables) for each study year. We accommodated NHIS complex sampling design and uneven probability of sampling among respondents by using population weights to compute all our summary and regression estimates, using the “svyset” command in Stata. We estimated change in prevalence of key mental health outcomes between baseline and each succeeding study year for each age subgroup and full sample by running linear regressions of each outcome on binary indicators for survey years (with 2019 serving as reference year), both before and after addition of controls. We also ran similar regressions for the summary variable for each age subgroup and used resultant estimates to evaluate the association of change in children’s mental health with known correlates of mental health. Finally, to track annual change in cumulative mental health burden among children, we plotted control-adjusted summary variable estimates for each group by study year. The University of Nevada, Las Vegas Institutional Review Board determined that this study did not require IRB review because it used existing, anonymized, publicly available data.

## 3. Results

Table 1 summarizes our overall study sample for each of the survey years. The incidence of most mental health problems in children initially rose from 2019 to 2020, then fell slightly back in 2021 without recovering fully, and finally, climbed sharply back again in 2022. With some exceptions, this pattern affected outcomes belonging to all domains of mental health but to varying extents. This pattern was particularly evident for ADD/ADHD; autism; learning disability; family members’ difficulty in understanding the preschool child; the difficulty of understanding school-age children by people inside or outside of the household; difficulty in learning things; anxiety; depression; and behavioral issues such as kicking/biting, controlling behavior, concentrating, and difficulty in playing and making friends. Table 1 also summarizes study covariates and provides strong evidence that our study samples retained a similar composition across survey years, with a few exceptions: food security rose noticeably and temporarily in 2021 compared to earlier years (possibly due to enhanced federal support), slightly more children were living in their own houses in 2021/22, parental employment dropped slightly over the study period before recovering in 2022, and the percentage of households with at least one elderly person rose slightly in 2020 and then remained steady.

In an unadjusted analysis of the full sample (Table 2), significant increases in parent-reported prevalence occurred between 2019 and 2022 for twelve of the nineteen outcomes we studied (ADHD/ADD, autism, learning disability, difficulty being understood by people inside and outside of the household, difficulty learning and remembering things, reported anxiety and depression at least weekly, difficulty controlling behavior, difficulty with changes in routine, and difficulty making friends). Most of these changes were sizable, ranging between 20 and 40% of the baseline prevalence. However, when we examine the regression result of individual age groups, most of the adverse changes appear to be heavily concentrated among adolescents (age group of 12–17). This group saw large increases in the prevalence of ADD/ADHD, autism, learning disabilities, difficulty in communicating with family members and outsiders, difficulty controlling behavior and concentrating, and difficulty learning and remembering things between 2019 and 2022. Adolescents also saw especially large increases in prevalence between 2019 and 2022 for three outcomes that did not show any worsening in 2020: reported anxiety and depression at least weekly and difficulty handling changes in routine and in making friends. For 6–11-year-old children, only two changes were significant between 2019 and 2022: learning disability and reporting anxiety at least weekly. No significant changes in prevalence for any mental health outcomes were found for preschool (2–5 years) children, with one exception: the prevalence of autism in this group increased by more than 130% between 2019 and 2022 (an increase in prevalence of 1.64 percentage points on a baseline of 1.47%). Adjustment for a slew of sociodemographic and other covariates yielded almost identical estimates for adolescents and 6–11-year-old children but with a slight reduction in effect sizes (Table 3). For preschool children, adjusted estimates for two items in addition to autism were significant: family members’ difficulty in understanding the child and difficulty learning things. Appendix A visually depicts the changes in the mean prevalence of all outcomes for the full sample of children.

Summary scores for mental distress (Figure 1), which track the trajectory of overall mental health between 2019 and 2022, suggest mental health for all children declined between 2019 and 2020 (1.01 to 1.18 points on a scale ranging from 0 to 15), then slightly improved but failed to recover fully in 2021 (1.09), and finally, continued to worsen in 2022 (1.25). Age subgroups show similar trajectories, but the decline was statistically significant for all three post-baseline years in adolescents only (1.11 at baseline, 1.24 in 2020, 1.30 in 2021, and 1.49 in 2022).

Appendix A summarizes the associations of pandemic-era changes in children’s mental health (captured by summary scores for cumulative mental distress) with several important known correlates of mental health in children. Female sex and belonging to a minority group seem to have exerted a strong protective effect on the mental health of children in all cohorts. Having public forms of insurance (compared to privately insured children) enhanced the risk of mental decline, while being uninsured seems to have exerted some protective effects. The family structure was predictive of mental health decline but only in adolescents: having two married parents at home was strongly protective compared to single-parent households. Parental employment and high food security were strongly associated with less mental health decline in all age sub-cohorts except preschool children. Better self-ratings of health were strongly associated with better mental health across all sub-cohorts. Bigger household size was mildly protective, but only in the 6–11-year age group. We did not find any association between mental distress and parental education, owning residences, or having an elderly person living in the household.

## 4. Discussion

Using nationally representative data from 2019–2022, we find that the advent and spread of COVID-19 coincided with sharp declines in the mental health of US children in 2020, which reversed slightly in 2021 before worsening further in 2022. These declines primarily affected adolescents and manifested as substantial increases in the prevalence of a broad set of indicators affecting major domains of mental well-being. Some researchers worry that the lack of representative data in the extant literature assessing pandemic-related impacts on child mental health may be painting an overly grim picture [6,7,18]. Our study helps address that concern by providing insights into population-level trends both before and during COVID-19. 

We found striking increases in the prevalence of three developmental disorders among adolescents: ADD/ADHD, autism, and learning disability. Our findings track several recent studies of ADHD, which report a global increase in ADHD symptoms during the pandemic [19,20,21,22,23,24,25]. For example, a recent meta-analysis found a small but significant increase in ADHD symptoms [19]. However, most studies do not provide data on US population prevalence. Our findings suggest the prevalence of ADD/ADHD among US adolescents jumped by roughly 25% between 2019 and 2022. Although it is unclear whether or how long such increases will be sustained into the future, their persistence into 2022, after substantial relaxation of COVID-19 mitigation measures, is of concern. For autism spectrum disorders, our findings paint a grimmer picture than some recently published studies [26,27]. For example, Wang et al. (2023) used the National Survey of Children’s Health to report a stable prevalence of parent-reported autism among 3–17 years old children between 2016 and 2021 [27]. In contrast, our findings suggest that the prevalence of autism increased significantly between 2019 and 2022 for both adolescents and preschool children. This was particularly notable for the latter group, in whom the prevalence jumped by nearly 130% from the 2019 baseline. Unlike previous studies of pandemic impacts, our study period covered the entire year of 2022, which may explain our starker findings. It should be noted that large increases in 2022 may at least partly result from post-pandemic recovery or the upsurge of developmental screening rates following declines during 2020–2021, leading to an underestimation of prevalence in the early pandemic period. While empirical data on screening rates during the pandemic is scarce, at least one study [28] did not find any significant erosion in developmental screening rates between 2019 and 2020. Collectively, our findings support expectations of a surge in demand for developmental health services that will require efficient allocation and targeting through existing and new treatment programs.

Our study substantiates concerns that symptoms of mood disorders (depression and anxiety) have become more prevalent among children during the pandemic, especially among adolescents [4,6,8,10,11,15]. We found large increases in depression and anxiety between 2019 and 2022 among adolescents and significant increases in anxiety among children aged 6–11 years, with most of the symptom surge happening in 2022. There is ongoing debate as to the extent to which childhood anxiety and depression have worsened in the wake of social disruptions caused by COVID-19’s spread and mitigation measures [7]. Some scholars have urged greater reliance on longitudinal cohort studies due to methodological biases afflicting cross-sectional studies [6,18]. High-quality cohort studies have yielded mixed results, and a number of meta-analyses summarizing the existing literature have been published [6,7,11]. A recent review of 53 longitudinal cohort studies from across the world reported a moderate increase in depression and anxiety, concentrated among high-income groups and females [6]. While longitudinal cohort (pre- and post-pandemic) data better captures changes in mental health, many cohort studies have small samples or non-representative sampling strategies that do not yield good information on the overall population prevalence. Further, published reviews of cohort data only included studies that were completed in 2020 and lacked the most recent 2022 data. Our findings of high levels of anxiety among school-age children, therefore, warrant attention and portend significant behavioral disruptions at home and in school that may require expansion of existing school-based mental health programs, especially those targeting high schoolers. 

The COVID-19 pandemic and mitigation measures aggravated social isolation, limited peer contact and play, and disrupted interpersonal school-based learning in school-going children. A growing body of literature suggests prolonged school closures impeded academic progress and contributed to “learning loss” in children [29,30]. However, there is a dearth of empirical data on the prevalence of pediatric cognitive problems. We find a significant worsening of learning and remembering ability among adolescents in 2022 relative to pre-pandemic levels. Adolescents were also more likely to face difficulty in communicating with people outside their households, have difficulty with making friends, and face challenges in handling changes in daily routine, all changes suggestive of erosion in social skills. Deficits in learning, memory, and communication may be a worrisome insignia of the intensity of social disruption engendered in children’s lives, and if persistent, could lead to impaired accumulation of human capital and long-term flourishing. Our findings are based on self-reports and should be considered preliminary. However, these results should prompt a more comprehensive examination of cognitive ability among children, using a more robust set of cognitive ability assessment scales and longitudinal cohort analysis.

We found that male and non-Hispanic white children faced steeper pandemic-era declines in overall mental health. Previous literature on race and gender disparities in mental health impacts has yielded mixed findings. Some studies have found a higher prevalence of affective disorders in girls [4,6], while there is some evidence that boys fared worse in coping with school closures during COVID-19 [30]. Non-Hispanic whites have been found to have a higher prevalence of mental health problems than similarly situated blacks or other minorities [31], a finding often attributed to better coping skills or resilience levels among minorities [32]. Few systemic reviews have evaluated measures of general mental well-being or assessed differential impacts by demographic subgroups. Instead, early work has focused heavily on mood disorders among adolescent children and found a higher incidence of sadness/depression in girls. While the focus on mood disorders is appropriate, future studies should track more global indices of mental health and employ sampling strategies that facilitate robust subgroup analysis. Consistent with earlier pre-pandemic literature, children living in bigger households [33], with employed parents [34], with both parents [35], and in food-secure environments [36] were less likely to see declines in mental health during the pandemic. Notable but temporary increases in household food security after a pandemic-driven surge in federal support payments in 2021 suggest that one avenue for mitigating the harmful impacts of the pandemic could be more efficient targeting of federal anti-poverty aid toward less food-secure households.

Our findings have several important implications for future research and policy. First, given the significant impact on children’s mental health, future studies should focus on apportioning the precise role of pandemic disease burden and the subsequent mitigation responses (e.g., lockdowns) on children’s mental wellbeing in order to provide more clarity on optimal policy responses in future pandemics. Second, since a large share of the burden of mental illness fell on adolescents, policymakers should explore ways to strengthen the existing system of behavioral screening and treatment available in middle and high schools; expand access to community-based behavioral health clinics that are integrated with primary care clinics, especially for communities facing shortages of behavioral healthcare providers; and facilitate a wider adoption of telehealth options. Finally, our findings support funneling more resources into aggressive screening programs for developmental disorders (like autism and ADHD) and cognitive deficits for school-going children that may have profound downstream impacts on their social adaption and long-term success.

## 5. Limitations

Our study has several limitations. First, we used data on repeated annual cross-sections of the US’s non-institutionalized population. Even though NHIS data methodology yields samples that are broadly representative, longitudinal panel data are more accurate at assessing the change in intensity or frequency of mental health symptoms across large intervals. However, there are inevitable tradeoffs in assembling large data sets that are both representative of national populations and capture change across multiple time points without non-random attrition. Obtaining precise estimates of the burden of mental health disease will likely require a judicious combination of both types of data. Second, our study is based on parent self-reports that are subject to recall and reporting biases. Third, our pre-pandemic baseline period for evaluating pandemic-related impacts is based on a single year of data (i.e., 2019), which is likely too short for distinguishing long-term pre-pandemic trends (i.e., secular trends) from pandemic-related impacts. Unfortunately, we were unable to use data prior to 2018 for capturing baseline trends because the NHIS made a major change in survey methodology after 2018 (in sampling, collection mode, and type of survey items) and warned researchers not to combine pre-2109 data with subsequent years. Finally, COVID-19 forced some changes in NHIS data collection in 2020–21, including a partial shift from in-person to telephone interviews, which led to a significant reduction in response rates compared to pre-pandemic periods. Reassuringly, we found very little change in the composition of key demographic groups between the 2019 baseline and the following years.

## 6. Conclusions

American children suffered a significant erosion of mental health against the backdrop of an evolving COVID-19 pandemic and the resultant society-wide spread mitigation responses. The declines swept broadly and persistently, affecting key domains of mental wellbeing well into 2022, with the burden falling especially heavily on adolescent children. Existing and new mental health programs, especially those aimed at school-going children, will likely need to be expanded to respond effectively to the ongoing crisis.

## Figures and Tables

**Figure 1 ijerph-21-00132-f001:**
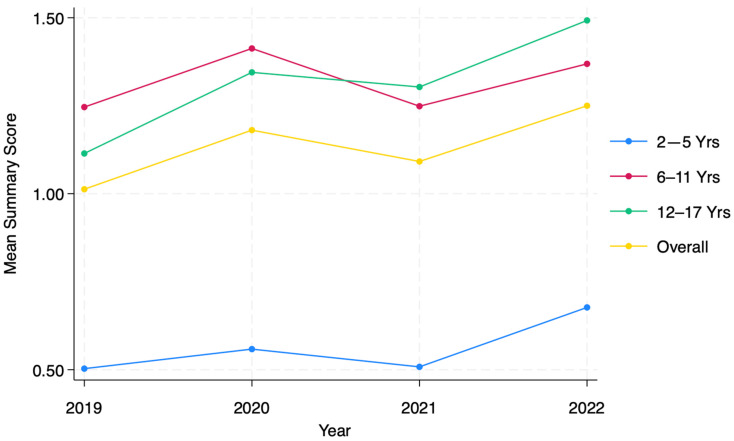
Summary scores of mental distress during study period of 2019–2022 by age group.

**Table 1 ijerph-21-00132-t001:** Summary statistics by study year.

Variables	Survey Year 2019 (N = 8188)		Survey Year 2020 (N = 5175)		Survey Year 2021 (N = 7382)		Survey Year 2022 (N = 6633)	
	Mean	SD	Mean	SD	Mean	SD	Mean	SD
**Outcomes**								
Developmental and learning disabilities								
*Currently has ADD/ADHD*	7.12	0.34	8.89	0.54	8.07	0.35	8.85	0.41
*Currently has intellectual disability*	1.19	0.15	1.53	0.22	1.18	0.15	1.35	0.17
*Currently has autism*	2.40	0.21	3.22	0.31	2.72	0.22	3.55	0.27
*Currently has developmental delay*	3.72	0.25	4.13	0.36	4.22	0.29	3.91	0.28
*Currently has learning disability*	5.34	0.29	6.71	0.44	5.96	0.34	6.91	0.39
Communication								
*Has difficulty understanding respondent*	2.79	0.58	2.69	0.62	3.74	0.69	3.97	0.66
*Family member has difficulty understanding child*	14.28	1.06	17.73	1.81	16.96	1.30	17.46	1.36
*Difficulty being understood by people inside household*	3.43	0.28	4.47	0.45	3.73	0.30	4.74	0.35
*Difficulty being understood by people outside household*	5.50	0.37	6.68	0.52	5.62	0.35	7.13	0.43
Cognition								
*Difficulty learning things*	8.14	0.38	9.54	0.55	8.97	0.38	9.33	0.43
*Difficulty remembering things*	7.55	0.43	7.91	0.55	8.00	0.40	8.80	0.45
Mood								
*Seems anxious, nervous, or worried at least weekly*	13.47	0.52	15.20	0.74	14.84	0.56	19.02	0.67
*Seems sad or depressed at least weekly*	5.21	0.30	5.99	0.57	5.10	0.32	7.03	0.41
Behavior								
*Difficulty playing*	0.23	0.13	0.44	0.25	0.12	0.09	0.93	0.39
*Kick, bite, or hit*	2.79	0.51	3.15	0.84	3.12	0.64	3.56	0.67
*Difficulty controlling behavior*	14.80	0.56	16.63	0.77	14.09	0.53	15.93	0.61
*Difficulty concentrating*	7.13	0.40	7.76	0.58	6.69	0.43	7.94	0.43
*Difficulty with changes in routine*	19.71	0.63	21.63	0.85	22.33	0.67	22.87	0.70
*Difficulty making friends*	8.92	0.42	11.02	0.64	9.76	0.45	11.60	0.49
Summary Score	1.01	0.03	1.18	0.04	1.09	0.03	1.25	0.03
**Controls/Covariates**								
Age (years)	9.54	0.06	9.62	0.08	9.62	0.06	9.73	0.07
Female sex	49.11	0.65	48.73	0.89	48.92	0.69	48.87	0.73
Race								
*White (reference)*	51.59	1.05	52.21	1.32	51.02	1.05	50.71	1.07
*Black*	12.83	0.67	12.91	0.89	12.31	0.65	12.46	0.66
*Hispanic*	25.76	1.05	25.50	1.30	25.71	1.00	26.06	1.07
*Asian*	4.31	0.29	4.30	0.32	4.66	0.29	4.57	0.29
*American or Indian native*	1.58	0.38	1.32	0.20	1.88	0.27	1.94	0.32
*Other*	3.93	0.29	3.77	0.38	4.41	0.29	4.25	0.31
Household size	4.24	0.01	4.23	0.02	4.23	0.01	4.26	0.02
Good or better self-rated health	97.15	0.24	97.70	0.29	97.92	0.21	97.26	0.24
Insurance coverage								
*Private (reference)*	55.99	0.91	55.57	1.12	55.18	0.89	54.99	0.89
*Medicaid and Other public*	35.98	0.86	35.81	1.13	37.52	0.89	37.16	0.89
*Other*	2.96	0.28	3.25	0.47	3.09	0.31	3.57	0.39
*Uninsured*	5.07	0.35	5.37	0.45	4.20	0.29	4.28	0.31
Family structure								
*Single parent (reference)*	29.12	0.70	28.65	0.90	28.55	0.70	27.40	0.72
*Married—two parents*	60.07	0.82	60.44	1.04	61.28	0.78	60.77	0.82
*Cohabitating—two parents*	6.45	0.38	6.36	0.50	6.33	0.35	6.45	0.37
*No parents*	4.36	0.28	4.56	0.40	3.84	0.26	5.39	0.33
Parental age (first residential parent)	39.72	0.12	39.90	0.15	40.13	0.12	40.30	0.01
Family race composition								
*Same-race family (reference)*	81.01	0.67	84.97	0.80	80.62	0.64	79.63	0.71
*Mixed-race family*	6.22	0.31	4.88	0.41	6.94	0.35	7.61	0.39
*Unknown or no parents*	12.77	0.61	10.14	0.70	12.44	0.60	12.76	0.65
Parental employment	78.05	0.64	75.04	0.95	75.72	0.69	78.21	0.69
At least one parent has a college degree	44.26	0.92	46.40	1.07	47.00	0.87	47.71	0.96
Federal poverty line-based family income categories								
0.0–0.49 *(reference)*	4.71	0.39	3.53	0.44	4.69	0.36	5.10	0.41
0.50–0.74	5.69	0.38	4.69	0.46	4.66	0.32	4.01	0.31
0.74–0.99	6.96	0.38	6.96	0.60	6.88	0.44	5.79	0.37
1.00–1.24	5.73	0.32	6.27	0.51	5.37	0.34	6.48	0.39
1.25–1.49	6.80	0.39	6.41	0.46	6.87	0.39	5.90	0.35
1.50–1.74	5.25	0.31	6.05	0.50	5.59	0.36	4.93	0.35
1.75–1.99	5.40	0.29	5.26	0.42	5.42	0.31	4.92	0.34
2.00–2.49	8.93	0.39	8.40	0.51	8.17	0.39	9.33	0.43
2.50–2.99	8.15	0.38	8.51	0.48	7.51	0.35	8.16	0.40
3.00–3.49	6.93	0.34	6.96	0.44	6.73	0.36	6.72	0.36
3.50–3.99	6.16	0.33	5.90	0.38	6.44	0.33	5.10	0.29
4.00–4.49	4.33	0.27	4.17	0.35	4.13	0.27	5.32	0.34
4.50–4.99	4.01	0.23	4.84	0.39	5.34	0.30	4.65	0.29
5.00 or greater	20.95	0.67	22.04	0.80	22.20	0.66	23.58	0.71
High food security	80.38	0.69	80.69	0.82	86.17	0.55	81.01	0.67
Owned residence	63.20	0.85	62.60	1.05	65.30	0.89	66.50	0.89
At least one elderly person lives in house	5.94	0.31	7.32	0.43	6.29	0.31	6.93	0.37

Notes: Child and parental age are shown in years, while the household size is shown as number of persons. All other variable means are shown as percentages.

**Table 2 ijerph-21-00132-t002:** Baseline (2019) prevalence and unadjusted change in prevalence of mental health outcomes by age group, National Health Interview Survey 2019–22.

	Full Sample (N = 27,378)				2–5 Years (N = 6222)				6–11 Years (N = 9298)				12–17 Years (N = 11,858)			
Mental Health Outcomes	Baseline Mean (2019)	Change			Baseline Mean (2019)	Change			Baseline Mean (2019)	Change			Baseline Mean (2019)	Change		
		2020	2021	2022		2020	2021	2022		2020	2021	2022		2020	2021	2022
Developmental and Learning Disabilities																
*Currently has ADD/ADHD*	7.12	**1.76 ***	0.94	**1.72 ***	0.91	0.07	0.21	0.02	8.60	1.25	0.01	0.82	9.68	**3.16 ***	**2.17 ***	**3.38 ***
*Currently has intellectual disability*	1.19	0.34	−0.01	0.16	0.86	−0.46	−0.53	0.40	1.18	0.71	0.08	0.13	1.42	0.47	0.21	0.03
*Currently has autism*	2.40	**0.82 ***	0.32	**1.15 ***	1.47	1.00	0.44	**1.64 ***	3.16	0.01	−0.05	0.57	2.27	**1.48 ***	0.58	**1.37 ***
*Currently has developmental delay*	3.72	0.41	0.49	0.18	4.62	0.05	0.74	0.31	3.98	−0.34	0.35	0.30	2.91	**1.35 ***	0.50	0.04
*Currently has learning disability*	5.34	**1.37 ***	0.62	**1.57 ***	1.98	0.33	0.11	0.56	5.89	1.50	0.82	**1.84 ***	6.96	**1.82 ***	0.69	**1.81 ***
Communication																
*Has difficulty understanding respondent*					2.79	−0.10	0.95	1.18								
*Family member has difficulty understanding child*					14.28	3.46	2.68	3.19								
*Difficulty being understood by people inside household*	3.43	**1.03 ***	0.30	**1.31 ***					3.92	1.70	0.04	0.96	2.62	0.46	0.79	0.99
*Difficulty being understood by people outside household*	5.50	1.18	0.12	**1.63 ***					6.56	**2.21 ***	−0.39	0.78	3.57	0.72	0.46	**2.10 ***
Cognition																
*Difficulty learning things*	8.14	**1.40 ***	0.83	**1.19 ***	4.93	0.19	−0.24	1.24	9.97	1.56	0.22	−0.05	8.43	**1.96 ***	**2.04 ***	**2.24 ***
*Difficulty remembering things*	7.55	0.36	0.45	**1.26 ***					8.56	0.12	−0.67	−0.12	7.10	0.70	1.63	**2.11 ***
Mood																
*Seems anxious, nervous, or worried at least weekly*	13.47	**1.73 ***	1.37	**5.54 ***					12.74	1.92	0.98	**3.89 ***	14.51	1.72	**2.40 ***	**8.34 ***
*Seems sad or depressed at least weekly*	5.21	0.78	−0.11	**1.82 ***					4.13	0.48	−0.56	1.19	6.74	0.51	0.40	**2.31**
Behavior																
*Difficulty playing*					0.23	0.21	−0.11	0.71								
*Kick, bite, or hit*					2.79	0.36	0.33	0.77								
*Difficulty controlling behavior*	14.80	**1.83 ***	−0.71	1.13					18.12	1.71	−1.45	−0.55	11.47	**2.68 ***	0.56	**2.41 ***
*Difficulty concentrating*	7.13	0.63	−0.44	0.81					8.99	−0.70	**−1.86 ***	−0.16	5.45	**2.41 ***	1.12	**1.85 ***
*Difficulty with changes in routine*	19.71	1.92	**2.62 ***	**3.16 ***					20.67	2.92	2.39	1.74	18.08	1.72	**3.68 ***	**4.84 ***
*Difficulty making friends*	8.92	**2.10 ***	0.84	**2.68 ***					8.24	1.72	0.34	0.96	10.40	1.88	1.61	**4.03 ***
Summary Score	1.01	**0.17 ***	**0.08 ***	**0.24 ***	0.50	0.06	0.01	**0.17 ***	1.25	0.17	0.00	0.12	1.11	**0.23 ***	**0.19 ***	**0.38 ***

Notes: (1) Single asterisk (*) indicates statistical significance at *p* < 0.05; (2) Except for summary score, baseline means are in percentages while change is displayed in percentage points; (3) Summary scores were unweighted sums of responses to binary survey items. Summary score maximums varied by age subgroup depending on the total number of survey items asked of respondents belonging to each subgroup; (4) Empty cells denote that the survey question was not asked of respondents belonging to the specific age group.

**Table 3 ijerph-21-00132-t003:** Baseline (2019) prevalence and adjusted change in prevalence of mental health outcomes by age group, National Health Interview Survey 2019–22.

	Full Sample (N = 27,378)				2–5 Years (N = 6222)				6–11 Years (N = 9298)				12–17 Years (N = 11,858)			
Mental Health Outcomes	Baseline Mean (2019)	Change			Baseline Mean (2019)	Change			Baseline Mean (2019)	Change			Baseline Mean (2019)	Change		
		2020	2021	2022		2020	2021	2022		2020	2021	2022		2020	2021	2022
Developmental and Learning Disabilities																
*Currently has ADD/ADHD*	7.12	**1.44 ***	0.86	**1.34 ***	0.91	0.04	0.16	−0.04	8.60	1.71	0.09	0.84	9.68	**2.16 ***	**2.08 ***	**2.79 ***
*Currently has intellectual disability*	1.19	0.39	0.00	0.14	0.86	−0.42	−0.55	0.35	1.18	0.68	0.11	0.12	1.42	0.56	0.16	−0.05
*Currently has autism*	2.40	**0.87 ***	0.32	**1.19 ***	1.47	**1.21 ***	0.60	**1.71 ***	3.16	0.03	0.05	0.74	2.27	**1.39 ***	0.32	**1.15 ***
*Currently has developmental delay*	3.72	0.41	0.63	0.06	4.62	0.71	1.24	0.80	3.98	−0.57	0.50	0.07	2.91	1.03	0.31	−0.48
*Currently has learning disability*	5.34	**1.33 ***	0.82	**1.41 ***	1.98	0.69	0.48	0.63	5.89	1.75	1.22	**1.78 ***	6.96	1.31	0.60	1.46
Communication																
*Has difficulty understanding respondent*					2.79	0.09	1.19	0.95								
*Family member has difficulty understanding child*					14.28	3.82	**3.85 ***	**3.76 ***								
*Difficulty being understood by people inside household*	3.43	**1.05 ***	0.46	**1.10 ***					3.92	**1.92 ***	0.34	0.93	2.62	0.23	0.69	0.41
*Difficulty being understood by people outside household*	5.50	1.22	0.40	**1.45 ***					6.56	**2.33 ***	0.03	0.73	3.57	0.64	0.59	**1.70 ***
Cognition																
*Difficulty learning things*	8.14	**1.41 ***	**1.20 ***	**1.22 ***	4.93	0.93	0.61	**1.81 ***	9.97	1.74	0.55	−0.29	8.43	1.36	**2.19 ***	**2.19 ***
*Difficulty remembering things*	7.55	0.51	0.85	**1.22***					8.56	0.29	−0.19	−0.15	7.10	0.45	**1.77 ***	**1.79 ***
Mood																
*Seems anxious, nervous, or worried at least weekly*	13.47	**1.69 ***	**1.69 ***	**5.02 ***					12.74	2.09	1.26	**3.44 ***	14.51	1.18	**2.59 ***	**7.53 ***
*Seems sad or depressed at least weekly*	5.21	0.86	0.27	**1.75 ***					4.13	0.67	−0.10	1.23	6.74	0.17	0.61	**1.80 ***
Behavior																
*Difficulty playing*					0.23	0.26	−0.06	0.75								
*Kick, bite or hit*					2.79	0.63	0.77	0.81								
*Difficulty controlling behavior*	14.80	**1.95 ***	−0.09	1.08					18.12	2.25	−0.61	−0.28	11.47	2.26	0.86	1.77
*Difficulty concentrating*	7.13	0.71	−0.13	0.62					8.99	−0.38	−1.46	−0.36	5.45	**2.04 ***	1.20	**1.45 ***
*Difficulty with changes in routine*	19.71	1.67	**3.02 ***	**3.12 ***					20.67	3.09	2.67	2.01	18.08	0.69	**3.91 ***	**4.21 ***
*Difficulty making friends*	8.92	**2.12 ***	**1.19 ***	**2.67 ***					8.24	1.75	0.55	0.80	10.40	1.64	**1.79 ***	**3.96 ***
Summary Score	1.01	**0.16 ***	**0.11 ***	**0.22 ***	0.50	0.07	0.04	**0.17 ***	1.25	**0.19 ***	0.05	0.12	1.11	**0.17 ***	**0.20 ***	**0.32 ***

Notes: (1) Single asterisk (*) indicates statistical significance at *p* < 0.05; (2) Except for summary score, baseline means are in percentages, while change is displayed in percentage points; (3) Summary scores were unweighted sums of responses to binary survey items. Summary score maximums varied by age subgroup depending on the total number of survey items asked of respondents belonging to each subgroup; (4) Empty cell denotes that the survey question was not asked of respondents belonging to the specific age group.

## Data Availability

NHIS data are publicly available at the following weblink: https://www.cdc.gov/nchs/nhis/index.htm, accessed on 26 November 2023.

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
