# Peer review of "Trends in Mental Wellbeing of US Children, 2019–2022: Erosion of Mental Health Continued in 2022"

_ijerph, 2024, doi:10.3390/ijerph21020132_

Round 1

Reviewer 1 Report

Comments and Suggestions for Authors

The article deals with a topic of current interest and need. It has several strengths such as the longitudinal nature of the study and the sample size. Some issues can be addressed to improve the quality of the work:

Introduction:

- It would be interesting to add more information on longitudinal studies in relation to the consequences that the pandemic has had on the health of young people. This would allow the reader to get a better idea of the background before focusing on the current study. 

Methodology:

- How is data collection and sampling carried out, what type of instrument was used online survey, paper and pencil, interviews? More information on the procedure is needed, it is not clear how the information is collected. 

- The outcome variables are not well specified. Were they assessed by diagnoses or by asking parents? Explain better. 

- Need to clarify if it has the approval of ethics committees. 

Results

- Adjust the tables to the formats required by the journal. 

- Some tables could be better explained with graphs that help to visually understand the results. 

Discussion

- Could be improved by including a paragraph on implications and recommendations taking into account the results.

Comments on the Quality of English Language

No comments

Reviewer 2 Report

Comments and Suggestions for Authors

I have read this paper with great interest, and value the effort.

In essence, this analysis provides additional support to quantify the impact of COVID and its management on mental health aspects in children.

In this way, these are useful to learn on how to manage future epidemics.

However, there are aspects that need to be somewhat more considered to improve the paper;

1.

We need some more information on the annual questionnaire.

Is this (on timing) at random over the year, or a given ‘time’ point, and are the same ‘cases’ recruited for consecutive years ?

Furthermore, it seems that only ‘health care seeking behavior’ is collected, although it seems that the individual themselves were collected, what is covered by ‘other related topics‘ (the 2.2. section on outcome variables suggest that also parental reporting is included ?), and how (2.2.) were these selected (while others likely not). Was this based on ‘consensus’, similar previous efforts or otherwise ?

2.

How does this 2019-2022 analysis ‘fits’ to previous reports, as ‘trend analysis’ is currently rather an ‘impact’ analysis.

The current analysis highly focuses on the years analysis, and hereby somewhat omit underlying patterns over a broader range, perhaps there are ‘similar’ fluctuations over time ? To rephrase my concern, the phenotypic differences cover both COVID and ‘generational’ trends, so that this analysis has limitations, or at least, some more clarity can be provided on this ‘phenotypic’ analysis.

3.

I understand the control variables concept, but some of these variables have perhaps also changed associated with the COVID period ? So, how ‘valuable’ or useful are these ?

Round 2

Reviewer 1 Report

Comments and Suggestions for Authors

The authors have addressed several of the issues suggested. However, I believe that in order to facilitate the interpretation of the results, the inclusion of graphs would be interesting. 

Reviewer 2 Report

Comments and Suggestions for Authors

the authors have addressed well my comments and reflections, that were to a certain extent similar to the comments of the other reviewer involved. I don't have anything to add to this. 
